# Ability of Garlic and Ginger Oil to Reduce *Salmonella* in Post-Harvest Poultry

**DOI:** 10.3390/ani12212974

**Published:** 2022-10-29

**Authors:** Kelsy Robinson, Anna L. F. V. Assumpcao, Komala Arsi, Annie Donoghue, Palmy R. R. Jesudhasan

**Affiliations:** 1Poultry Research Unit, ARS, USDA, Mississippi State, MS 39762, USA; 2Department of Poultry Science, University of Arkansas, Fayetteville, AR 72701, USA; 3Poultry Production and Product Safety Research Unit, ARS, USDA, Fayetteville, AR 72701, USA

**Keywords:** *Salmonella*, chicken, poultry, garlic, ginger, phytochemical, antibiotic alternatives

## Abstract

**Simple Summary:**

Reducing *Salmonella* contamination of poultry meat is vital to reducing the incidence of human Salmonella infections in the United States. While individual birds can carry *Salmonella,* poultry meat often becomes contaminated in the processing plant where a high risk of cross contamination also exists. Currently, chemicals such as peracetic acid are used during processing to kill microbes and prevent contamination. However, consumers are increasingly weary of chemical use with interest in natural alternatives growing. Garlic and ginger oils have proven antimicrobial activity and possess many benefits for human health. Therefore, we sought to determine the ability of garlic and ginger to reduce Salmonella in the processing environment. A combination of both oils reduced *Salmonella* on chicken skin by up to 99% and prevented *Salmonella* growth in the simulated scalding tank environment. Investigation into the antibacterial mode-of-action revealed a potential of the oils to inhibit the ability of *Salmonella* to adhere to host cells. In total, this data demonstrates the potential of garlic and ginger oil to serve as a natural alternative to reduce *Salmonella* prevalence and cross-contamination in the processing plant.

**Abstract:**

Approximately 1.35 million human salmonellosis cases are reported in the United States every year, resulting in over 26,000 hospitalizations and 400 deaths. Consumption of contaminated poultry products is one of the leading causes of human salmonellosis. Poultry meat becomes contaminated when feces from an infected bird comes into contact with the carcass during processing. Additional carcasses can then become cross-contaminated along the processing line. While chemicals such as peracetic acid are currently used to kill microbes such as *Salmonella*, consumers are increasingly calling for more natural alternatives. Our objective for this study was to determine the ability of the phytochemicals garlic and ginger oil to reduce *Salmonella* prevalence in the processing environment. In a simulated scalding tank environment, dipping contaminated chicken skin samples in a solution containing both garlic and ginger oil reduced Salmonella by up to 2 log CFU. Furthermore, the oils prevented *Salmonella* growth in the tank solution. The mechanism of action of garlic and ginger was evaluated using the sub-inhibitory concentration of each oil individually. While both were found to decrease autoinducer-2 (AI-2) levels, no effect was seen on expression of 10 genes involved in *Salmonella* virulence and survival. In total, this work demonstrates the potential of garlic and ginger to reduce *Salmonella* prevalence in the post-harvest environment. However, more work remains to be done to understand the mechanism of action.

## 1. Introduction

Salmonellosis is a human gastroenteritis caused by the rod-shaped, Gram-negative bacterium *Salmonella*. Approximately 1.35 million human salmonellosis cases are reported annually in the United States, resulting in over 26,000 hospitalizations and 400 deaths [1]. Infections are commonly caused by the consumption of raw or undercooked food. While several sources have been identified, contaminated poultry products are the primary source accounting for approximately 10–29% of *Salmonella* cases [2].

Chickens serve as a natural host for *Salmonella,* which can colonize almost every part of the intestinal tract. Colonization often occurs early in life via vertical transmission or environmental exposure at the hatchery, during transportation, or at the farm [2,3]. Infected birds can then spread *Salmonella* horizontally during grow-out, with transmission often associated with stressful events such as feed withdrawal prior to slaughter [4]. During slaughter and evisceration, carcasses can be exposed to fecal material, causing the meat to become contaminated with *Salmonella*. Additional carcasses often become cross-contaminated during processing creating a significant public health risk [5].

Control of *Salmonella* in poultry currently involves both pre- and post-harvest interventions. Pre-harvest biosafety measures such as disinfection of eggs and the hatchery environment and fomite and vector control aim to prevent transmission and colonization [2,6]. However, the ubiquitous nature of *Salmonella* serovars limits the effectiveness of these measures. Antibiotics have been used to decrease intestinal load and shedding but are associated with the development of antimicrobial resistance in bacterial pathogens [7,8,9]. This has made antibiotic use in livestock a significant public health concern and created a need for safer antibiotic alternatives. During post-harvest, peracetic acid (PAA) is used to disinfect commercial poultry products [10]. However, consumer acceptance of PAA diminishes with an increased preference for natural alternatives.

Phytochemicals have recently emerged as a novel antibiotic alternative with potential for use in both pre- and post-harvest settings. They are capable of killing bacteria through disruption of the cell wall and membrane and can modulate bacterial virulence [11]. Dietary inclusion of some phytochemicals has been shown to improve the innate immune response of chickens, promote growth and exert beneficial effects on gut health [12,13,14]. Additionally, several studies have investigated the ability of phytochemicals to serve as antimicrobial washes and coatings against *Campylobacter jejuni* [15,16,17,18,19,20,21,22].

Garlic (*Allium sativum*) has been used as a spice in every diet worldwide since ancient times [23]. However, it’s benefits extend beyond flavor to multiple health benefits such as anti-cancer, anti-inflammatory, antifungal, and antiviral activities [24]. Garlic has also been used to treat bacterial infections for centuries [24]. Susceptibility to crude garlic extract has been shown for the bacteria *Escherichia coli*, *Pseudomonas aeruginosa*, and *Proteus* sp. [25]. The organosulfur compounds allicin, ajoene, and various aliphatic sulfides have been identified as the primary components responsible for garlic’s antibacterial activity [24], which are also found in garlic oil [25,26] and has been shown to have antibacterial activity against *Mycobacterium tuberculosis* [27,28], *Helicobacter pyori* [29], *Staphylococcus aureus* [30,31], *Bacillus cereus* [32], *E. coli* [32,33], *Shigella* species [32], *Vibrio* species [32], *Yersinia enterocolitica* [32], *Pseudomonas aeruginosa* [31,34], *Klebsiella pneumoniae* [34], *Campylobacter jejuni* [32,33,35], *Salmonella typhimurium* [32,33], and *Listeria monocytogenes* [32].

Similar to garlic, ginger is a spice that has been used in cooking and traditional medicine in several cultures worldwide. Ginger has several active compounds, such as gingerol, gingerdiol, shogaols, gingerdione, and zingerones [36]. It has been shown to have multiple human health benefits, including antibacterial activity. The antibacterial activity of ginger has been tested against multiple bacterial strains with many of the strains, including *Salmonella*, found to be susceptible [37,38,39].

Previously, we demonstrated the ability of garlic and ginger oil to damage/kill the foodborne pathogen *Campylobacter jejuni*, inhibit its adhesion to chicken embryo cells, and reduce its quorum sensing [10]. Therefore, the goal of this study was to further determine the suitability of these oils as antimicrobials for poultry production by investigating their antimicrobial activity individually and the combination of both against three *Salmonella* serovars (*S. enteritidis*, *S. typhimurium*, and *S. infantis*) and to determine their possible mode of action.

## 2. Materials and Methods

### 2.1. Salmonella Strains and Culture Conditions

Three strains of *Salmonella* were used for this study (*S. infantis* BAA 1675, *S. enteritidis* PT4 NCTC 13349 nalidixic acid (NA) resistant, *S. typhimurium* ATCC 14028 NA resistant). Bacterial strains were separately cultured in tubes containing tryptic soy broth [TSB] (Hardy Diagnostics, Santa Barbara, CA, USA) and incubated overnight at 37 °C.

### 2.2. Preparation of Garlic and Ginger Solutions

Garlic and ginger oils were purchased from Sigma-Aldrich Co. (St. Louis, MO, USA). A 10% stock solution of both oils was created by diluting in 90% ethanol. The stock solutions were further diluted in TSB to obtain the desired concentrations for each subsequent experiment.

### 2.3. Antimicrobial Activity against S. infantis on Chicken Skin

The antimicrobial activity of garlic and ginger oil against *Salmonella* on chicken skin was determined according to a previously published protocol [10,21]. A total of three independent experiments were conducted. Briefly, chicken skin samples were obtained from a local grocery store, aseptically cut into 4 cm × 4 cm pieces, and exposed to UV light for 15 min to ensure they were free of viable bacteria. Skin samples were randomly divided into 10 groups, with three pieces per group. Experimental groups included baseline (not dipped in solution), positive control (Butterfield’s Phosphate Diluent), ethanol alone, 0.5% garlic oil, 1.0% garlic oil, 0.5% ginger oil, 1.0% ginger oil, 0.5% garlic oil with 0.5% ginger oil, and 1.0% garlic oil with 1.0% ginger oil. Baseline samples were included to measure bacterial adherence to the skin, while positive control samples were dipped in Butterfield’s Phosphate Diluent to account for the bacteria that were washed off during submersion. For the ethanol group, ethanol was included at the same concentration found in the 1.0% garlic and ginger oil treatment (9.0% ethanol). Skin samples were inoculated with 50 μL of *Salmonella* Infantis (~8.5 to 9 CFU/mL). Following inoculation, samples were incubated for 30 min to ensure adherence of bacteria to skin samples. Inoculated samples were immersed in the respective treatment solutions for either 30 s or two minutes at 54 °C shaking, and then immediately transferred to 5 mL of Dey-Engley neutralizing broth (Difco Laboratories, Sparks, MD, USA) and vigorously vortexed for 30 s. The neutralizing broth was serially diluted (1:10) and placed on xylose lysine deoxycholate (XLD) agar plates (HiMedia Laboratories, Thane West, Maharashtra, India). To determine the effect of treatments on bacterial growth in the wash solution, treatment solutions were plated on XLD plates at 0 and 2 h post-treatment. Plates were incubated at 37 °C for 24 h for *S. infantis* enumeration.

### 2.4. Determination of Sub-Inhibitory Concentration

The sub-inhibitory concentration (SIC) of garlic and ginger oil was determined according to a previously published protocol [10]. Two-fold dilutions of garlic and ginger oil (0, 2.5, 1.25, 0.625, 0.312, 0.156, 0.078, 0.039%) were prepared in a sterile 96-well plate (Costar, Corning, NY, USA) containing 100 μL of TSB broth. Ethanol was also included at the concentrations found in the corresponding garlic and ginger samples. Wells were subsequently inoculated with 100 μL of *S. infantis*, *S. typhimurium*, or *S. enteritidis* (~10^6^ CFU/mL). Plates were placed in a Cytation 5 multimode microplate reader (Agilent, Santa Clara, CA, USA) and incubated at 37 °C for 24 h with OD600 readings taken every 30 min. Following incubation, samples were diluted, plated onto XLD plates, and incubated for 24 h at 37 °C for *Salmonella* enumeration. All concentrations were run in duplicate, and experiments were repeated three times for each *Salmonella* strain. The highest concentration of phytochemicals that did not inhibit *Salmonella* growth was selected as the SIC and used in subsequent experiments.

### 2.5. Impact of Garlic and Ginger on Quorum Sensing Activity of Salmonella

The *Vibrio harveyi* bioluminescence assay was used to determine the effect of SIC of garlic and ginger on autoinducer-2 (AI-2) levels in *Salmonella* strains, as described previously [10,15]. Briefly, *S. infantis*, *S. typhimurium*, and *S. enteritidis* were cultured with or without SIC of garlic or ginger oil. Following incubation, cells were centrifuged and the supernatant was collected and filtered to obtain the cell-free supernatant (CFS). CFS was also obtained from *V.*
*harveyi* strain BB152 following overnight growth in Luria Bertani broth (HiMedia Laboratories Pvt. Ltd., Mumbai, India). *V.*
*harveyi* strain BB170, the reporter strain, was grown in Luria Bertani broth to ~3.5 log CFU/mL, diluted 1:5000 with autoinducer assay medium, and dispensed into 96-well microtiter plates. *V. harveyi* BB152 CFS (positive control), autoinducer assay medium (negative control), or treated or untreated *Salmonella* CFS were added to respective wells. Luminescence was measured in a Cytation 5 multimode microplate reader (Agilent) every 20 min for 8 h at 30 °C. Luminescence observed in the negative controls due to self-induction of *V. harveyi* BB170 was deducted from the positive controls and treatments before further analysis. All samples were run in duplicate on the plate, and the experiment was repeated one time.

### 2.6. Changes in Salmonella Gene Expression following Garlic and Ginger Exposure

Real-time PCR was used to measure the influence of garlic and ginger oil on the expression of critical *Salmonella* genes. *Salmonella* strains were grown to the mid-log phase in the presence or absence of SIC of garlic or ginger in TSB at 37 °C. The Quick-RNA Fungal/Bacterial Miniprep Kit (Zymo Research) was used to extract RNA, which was treated with DNase (Zymo Research) and subjected to reverse transcription using the iScript cDNA synthesis kit (Bio-Rad) according to the manufacturer’s protocol. Primers (Table 1) were designed from published GenBank *Salmonella* sequences using Primer 3 software (National Center for Biotechnology Information) and purchased from Integrated DNA Technologies. *16S* was used as an internal control for normalization, and quantification was determined by the delta-delta cycle threshold (ΔΔCt) method.

### 2.7. Statistical Analysis

*Salmonella* counts were logarithmically transformed before analysis to achieve homogeneity of variance [10]. All experiments were conducted as completely randomized designs with six replicates per treatment. Data from independent trials were pooled and analyzed using ANOVA in GraphPad Prism version 9.1 (GraphPad Software, San Diego, CA, USA). Tukey’s test was used for multiple comparisons among treatments in all assays. A *p*-value of <0.05 was considered statistically significant.

## 3. Results

### 3.1. Antimicrobial Activity of Garlic and Ginger Oil in the Scalding Tank Environment

The antimicrobial activity of garlic and ginger in the scalding tank environment was evaluated, as the scalding tank represents one of the major potential sites of *Salmonella* cross-contamination during poultry processing. In three independent experiments, contaminated skin samples were dipped into 56° C solutions containing 0.5% or 1% garlic and ginger alone and in combination for either 30 s or 2 min. Baseline results revealed the skin samples to have a bacterial load of approximately 6.5 log CFU/sample (Figure 1). Bacterial loss due to submersion was measured by dipping samples in BPD solution (positive control). Submersion for 30 s or 2 min resulted in similar levels of bacteria losses, with levels decreasing to 5.8 and 5.7 log CFU/sample, respectively. The presence of bacteria in these samples following submersion also confirms that the scalding tank temperature (56 °C) had no effect on bacterial viability. While no antimicrobial activity was observed for 0.5% garlic or ginger oil alone, immersion of samples in 1% of either oil resulted in a numerical decrease in *S. infantis* at 30 s and a significant decrease at 2 min. Antimicrobial activity increased when the oils were combined with 0.5% garlic and ginger oil together, resulting in a 0.8 log CFU/sample decrease in *S. infantis* at 30 s and a 1.2 log CFU/sample decrease at 2 min. Similar to both oils alone, antimicrobial activity was further enhanced when garlic and ginger oil were combined at 1% each resulting in a 1 and 2 log CFU/sample *Salmonella* reduction at 30 s and 2 min, respectively. Additionally, a comparison of the two submersion times revealed no significant difference between 0.5% ginger and garlic combination at 30 s and 2 min, while the 1% garlic and ginger combination was significantly different between time points (data not shown).

To evaluate the ability of garlic and ginger oil to prevent bacterial growth in the scalding tank environment, solutions were sampled and plated immediately following treatment and again at 2 h post-treatment. The presence and absence of *Salmonella* growth in each sample are shown in Table 2. All positive control samples containing only BPD were positive for *Salmonella* at both time points.

Solutions of 0.5% ginger or garlic oil alone had little to no effect on *Salmonella* levels immediately following treatment. However, they did show some antimicrobial activity after 2 h of incubation with only 7/15 and 5/15 positive samples, respectively. However, strong antimicrobial activity was observed for 1% garlic and ginger oil alone and in combination and the combination of 0.5% garlic and ginger oil. More than 86% of samples were negative for *Salmonella* immediately following treatment, while 100% were negative at 2 h post-treatment. It is important to note that the solution containing the same level of ethanol as the 1% garlic or ginger oil solutions reduced the number *Salmonella* positive samples. However, the effect at 0 h post-treatment was much weaker than the oil treatments indicating that the antimicrobial activity can be attributed to the oil rather than the presence of ethanol.

### 3.2. Effect of Various Concentrations of Garlic and Ginger Oil on Salmonella Growth

Inhibition of bacterial growth by different garlic and ginger oil concentrations was determined using three *Salmonella* strains. Inhibition of growth was measured by taking OD600 readings every 30 min for 24 h (Figure 2). To confirm the OD600 results, *S. infantis* samples were serially diluted and plated following the 24-h incubation (Appendix A). The inclusion of either oil at 0.312% resulted in an apparent reduction in the growth of all three strains. The presence of 0.156% of either oil had no statistical effect on the growth of any strain through a numerical decrease *S. enteritidis* was observed (Figure 3C,D). Interestingly, a slight increase in the growth of *S. infantis* and *S. typhimurium* was observed in the presence of 0.078% ginger oil (Figure 3A,C). Therefore, 0.156% was determined to be the SIC for both oils and was used for future mechanistic studies.

### 3.3. Inhibition of Quorum Sensing by Garlic and Ginger Oil

The effect of SIC of garlic and ginger oil on AI-2 levels is shown in Figure 3. Ethanol, included at levels equal to that found in the garlic and ginger oil SIC, increased AI-2 relative light units of *S. infantis* and *S. enteritidis* (Figure 3A,D) relative to untreated controls but had no effect on *S. typhimurium* (Figure 3G). Ginger oil significantly reduced AI-2 relative light units associated with *S. enteritidis* and *S. typhimurium* as compared to ethanol alone, beginning at 4 h of incubation and continuing through 8 h (Figure 3E,H). No effect was seen on AI-2 relative light units for *S. infantis* (Figure 3B). However, a significant reduction in AI-2 relative light units was observed for all three strains in the presence of garlic oil when compared to the ethanol control (Figure 3C,F,I). Significant decreases were observed as early 30 min and became consistent at 4 h of incubation, continuing through the end of the experiment.

### 3.4. Effect of Garlic and Ginger on Salmonella Gene Expression

The expression of ten genes involved in *Salmonella* virulence and survivability was determined following the incubation of *S. infantis*, *S. enteritidis*, and *S. typhimurium* with the SIC of garlic and ginger oil individually (Figure 4). No effect on any of the genes tested was observed in *S. infantis* (Figure 4A,B) or *S. typhimurium* (Figure 4E,F). Additionally, no significant difference in gene expression was observed for *S. enteritidis* in the presence of the SIC of ginger oil (Figure 4C). However, the SIC of garlic oil did significantly reduce the expression of invasion protein A (invA) in *S. enteritidis* (Figure 4D).

## 4. Discussion

Controlling *Salmonella* colonization and growth at all levels of poultry production is critical to reducing the number of human salmonellosis cases observed each year. Within the farm-to-fork pipeline, the processing plant is one of the most important control points as cross-contamination of carcasses at this point can create a serious public health concern [5]. While antibacterials such as PAA are currently used to eliminate food safety pathogens such as *Salmonella,* consumers are increasingly searching for effective natural alternatives [10]. Garlic and ginger oil have been used for medicinal purposes since ancient times due to their multiple health benefits [24,36]. They have been shown to have antimicrobial activity against Gram positive and Gram negative bacteria [32,33,36], are generally recognized as safe, and are considered to be environmentally friendly. We previously demonstrated the ability of these oils to reduce *Campylobacter jejuni* when used as a chill tank treatment [10]. However, immersion chilling is beginning to be replaced by other technology, such as air spray chilling. Therefore, we followed up on this previous work by determining the suitability of garlic and ginger oils as a scalding tank treatment.

Scalding tanks are an essential part of poultry processing as they loosen the feathers from the skin to make removal easier, but they also serve as one of the first potential sites of cross-contamination. Following euthanasia, carcasses are submerged in the tanks for 2.5 to 3.5 min. Fecal material on the carcasses can then contaminate the scalding tank water allowing harmful bacteria to come into contact with exposed meat. In this study, combining garlic and ginger oil at either 0.5% or 1% significantly reduced *Salmonella* on chicken skin over control. Specifically, the combination of 1% garlic and 1% ginger oil produced a substantial reduction in *Salmonella,* approximately 1 log (90%), and 2 log (99%) reduction, respectively, after 30 s and 2 min. These results are promising as a 50% reduction in the prevalence of *Salmonella-*contaminated poultry (from 20% to 10%) can produce a 50% reduction in the expected illnesses per serving, whereas a significant reduction in prevalence from 20% to 0.05% can reduce the risk of illnesses by 99.75% per serving [40]. Thus, a one-to-one relationship has been estimated, indicating that a reduction in the prevalence of *Salmonella-*contaminated chicken can reduce the risk of human infection by a similar percentage. Furthermore, plating the solutions immediately following treatment and 2 h post-treatment reduced the number of *Salmonella-*positive samples by 87% to 100% (Table 2).

While multiple studies have investigated the role of garlic and ginger in reducing *Salmonella* during rearing and their subsequent effect on bird performance and health [41,42,43,44,45,46,47], few have determined their post-harvest efficacy [48,49]. Similar to our results, Sudarchan et al. [48] reported a significant 0.40 and 0.13 log CFU reduction in *Salmonella* when contaminated chicken meat was dipped in a 0.66% solution of garlic and ginger oil, respectively. However, De Moura Oliveria and others [48] found that a 5% to 15% garlic solution was necessary to eliminate *Salmonella* recovery on stored, chilled carcasses. This is most likely due to the form of garlic used in the different studies, as aqueous garlic has been shown to be less effective than the oil extracts [49,50].

The SIC of garlic and ginger oil was used to determine the garlic and ginger antibacterial mode of action as it has been shown to change the virulence qualities of bacteria without killing the pathogen [10,21,22,51]. The molecule AI-2 is responsible for mediating quorum sensing in bacteria which is crucial for systems such as colonization, biofilm formation, motility, and interactions with host cells [15,52]. AI-2 is produced when the *luxS* gene cleaves S-ribosylhomocysteine into homocysteine and 4,5-dihydroxy-2,3-pentanedione, the latter of which undergoes spontaneous cyclization to form AI-2 [53]. Garlic oil was shown to significantly reduce AI-2 production in *S. infantis*, *S. enteritidis*, and *S. typhimurium*. In contrast, ginger oil had a better effect on *S. enteritidis* and *S. typhimurium* than *S. infantis* (Figure 3). To our knowledge, this is the first study to demonstrate the effect of these oils on *Salmonella* AI-2 production. However, they have been shown to reduce AI-2 in *Campylobacter jejuni* [10]. Taken together, this may indicate that garlic and ginger inhibit the interaction of *Salmonella* with chicken skin cells by interfering with AI-2 production. Finally, we evaluated the effect of the SIC of garlic and ginger oil on the expression of 10 virulence and survival genes. However, only garlic oil was shown to have an effect that was restricted to the expression of *invA* in *S. enteritidis*, indicating that it was not a universal effect. The genes chosen for this study were a subset of those responsible for *Salmonella* survival. Screening of additional genes may provide further information on the mechanism of action of garlic and ginger oil.

## 5. Conclusions

Reduce *Salmonella* in the post-harvest environment is essential to controlling the rate of human salmonellosis associated with poultry meat. Here, we show the ability of garlic and ginger oil alone and together to reduce *Salmonella* on contaminated poultry meat and prevent *Salmonella* growth when used as a scalding tank treatment. The up to 2 log reduction in *Salmonella* seen in this study is significant and demonstrates the potential of these oils to produce meaningful real-world reductions in human salmonellosis cases. Future studies should focus on the effect of these oils on meat quality and sensory characteristics while also investigating additional post-harvest uses.

## Figures and Tables

**Figure 1 animals-12-02974-f001:**
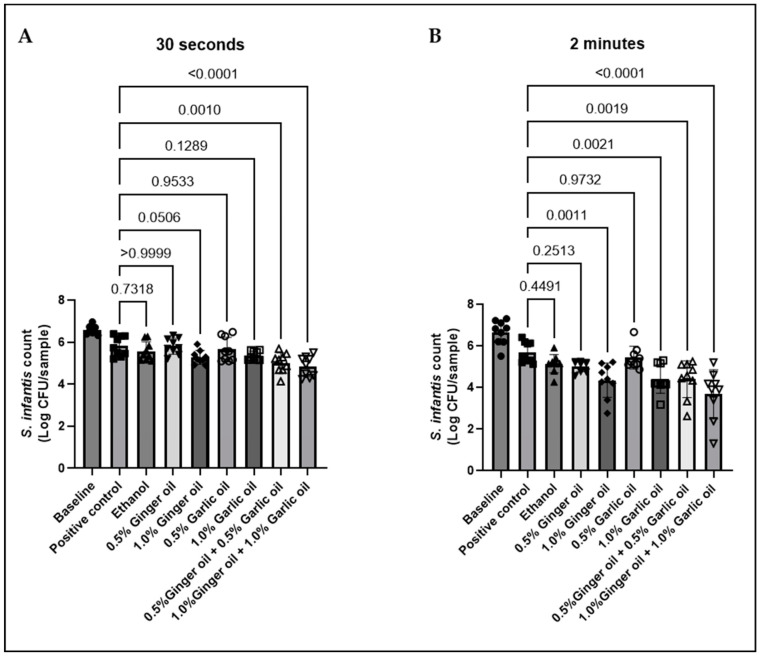
Chicken skin samples were inoculated with 7.55–8.1 Log CFU/sample of *S. infantis* and dipped in 56 °C solutions containing ginger or garlic oil alone or in combination for 30 s (**A**) or 2 min (**B**). Results are averages of three independent trials, each containing triplicate samples (mean and STD). The estimated *p*-values are included above the bars.

**Figure 2 animals-12-02974-f002:**
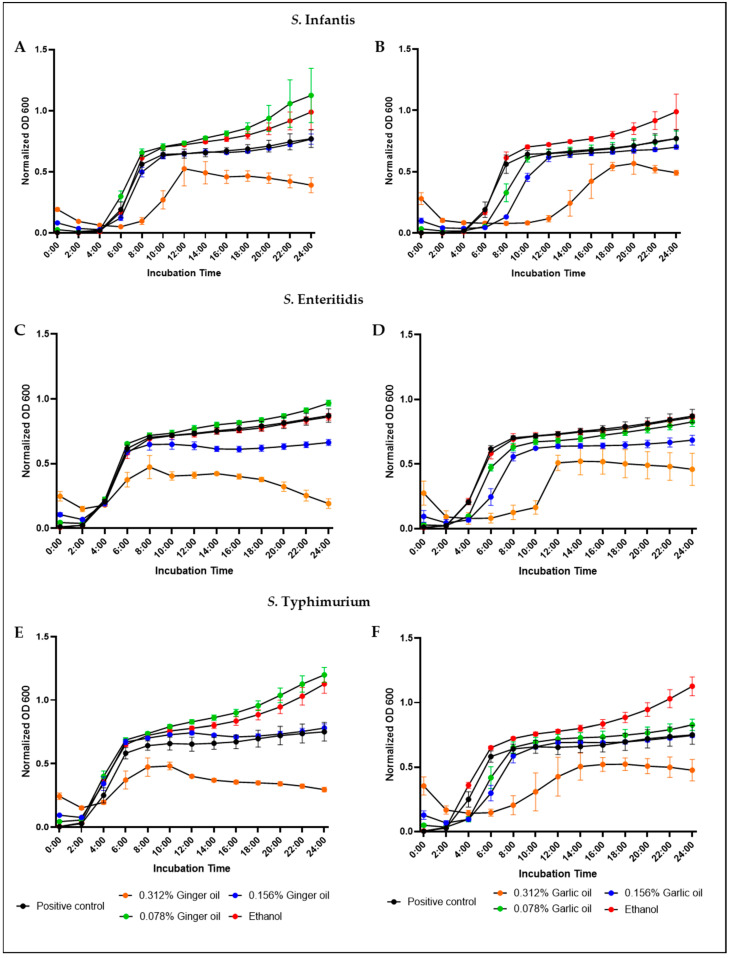
Determination of SIC of ethanol, ginger oil, and garlic oil against *S. infantis* (**A**,**B**), *S. enteritidis* (**C**,**D**), and *S. typhimurium* (**E**,**F**) in a microtiter plate. Results are averages of two independent experiments, each containing duplicate samples (mean and STD). Experiments using ginger oil are shown in the first column, and garlic oil in the second column.

**Figure 3 animals-12-02974-f003:**
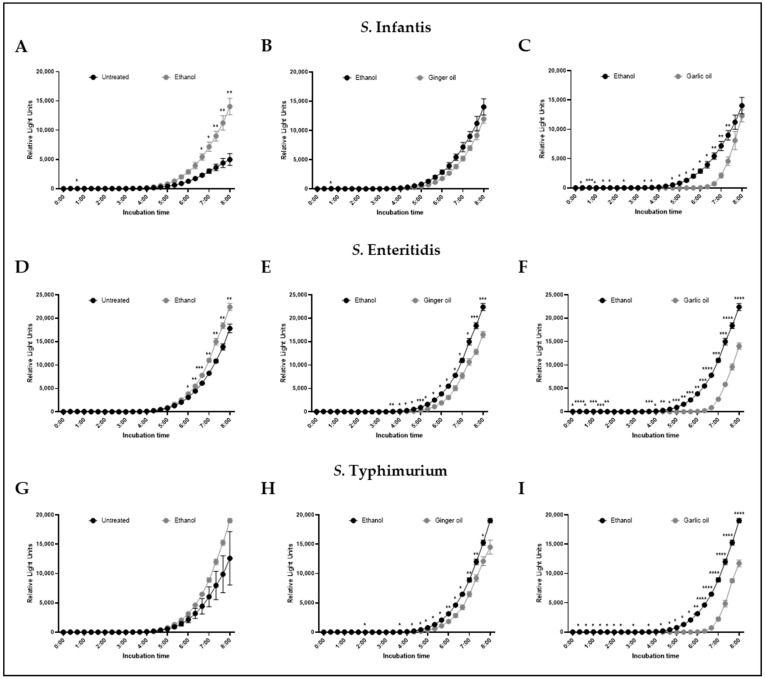
Effect of ethanol, ginger oil, and garlic oil at SIC dose on AI-2 levels of *S. infantis* (**A**–**C**), *S. enteritidis* (**D**–**F**), and *S. typhimurium* (**G**–**I**) determined by bioluminescence assay. Results are averages of two independent experiments, each containing duplicate samples (mean and STD). ** p <* 0.05; *** p <* 0.01; **** p <* 0.001; ***** p <* 0.0001.

**Figure 4 animals-12-02974-f004:**
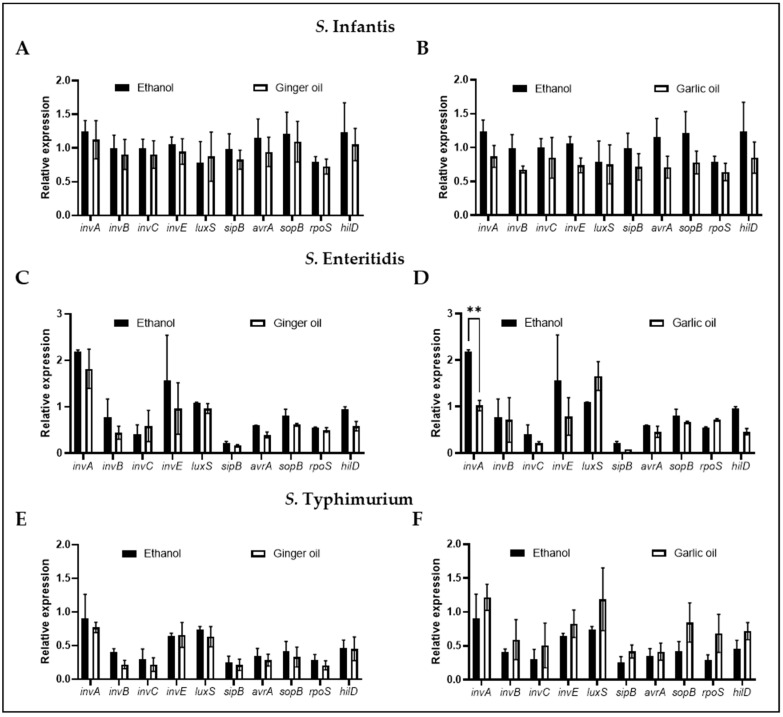
Effect of garlic and ginger oil on the expression of genes critical for survival and virulence was determined using qPCR. *S. infantis* (**A**,**B**), *S. enteritidis* (**C**,**D**), and *S. typhimurium* (**E**,**F**) (~6.0 Log CFU/mL) were incubated at 37 °C for 4 h in the presence of the SIC of phytochemicals. 16S-rRNA served as the endogenous control. Results are averages of four biological replications (mean and STD) and were considered significant at *p* < 0.05 (** *p* < 0.01).

**Table 1 animals-12-02974-t001:** Primers used for gene expression analysis using quantitative real-time PCR.

Gene with Accession No.	Product Size	Primer	Sequence (5′-3′)
16S-rRNA (NC_003197.2)	90 bp	F	5′-TTGTACACACCGCCCGTCAC-3′
		R	5′-AAAGTGGTAAGCGCCCTCCC-3′
invA (NC_003197.2)	94 bp	F	5′-CCGATTTGAAGGCCGGTATTA-3′
		R	5′-ACCGTCAAAGGAACCGTAAAG-3′
invB (NC_003197.2)	92 bp	F	5′-AAGTATCTGTATCAGCGTCAAGG-3′
		R	5′-TCATAAGCCCGCTGTTGTAATA-3′
invC (NC_003197.2)	103 bp	F	5′-GTTATCCCGCCTCCGTATTC-3′
		R	5′-GCTTTCCAGCAGTACCGTATAA-3′
invE (NC_003197.2)	82 bp	F	5′-GTCAGGCGCGTAGCTTATT-3′
		R	5′-CACGATCTCTTCCAGGTCTTTAC-3′
luxS (NC_003197.2)	122 bp	F	5′-TGCTGAAAGTGCAGGATCAA-3′
		R	5′-GCACATCACGCTCCAGAATA-3′
sipB (NC_003197.2)	121 bp	F	5′-CGACGGGAGTGTCGTTTATT-3′
		R	5′-CTTATCGACGCCTAATCCTTCC-3′
avrA (NC_003197.2)	135 bp	F	5′-GTCCATGAGCTTGTTTCCTCTA-3′
		R	5′-CCGATGTCTTTCCGTCCATAA-3′
sopB (NC_003197.2)	128 bp	F	5′-CACTCGCTGCATAACCTCTATAA-3′
		R	5′-GTCCGCTTTAACTTTGGCTAAC-3′
rpoS (AF184104.1)	103 bp	F	5′-GATAACGACCTGGCTGAAGAA-3′
		R	5′-ACAGTGGTGAATACCCAATCTC-3′
hilD (NC_003197.2)	97 bp	F	5′-GGCGCTCTCTATGCACTTATC-3′
		R	5′-GCAGGAAAGTCAGGCGTATAG-3′

**Table 2 animals-12-02974-t002:** Antimicrobial efficacy of phytochemicals as scalding tank treatment to reduce *S. infantis* in the treatment solutions after treatment and 2 h after treatment.

Treatment	*Salmonella*-Positive (Positive Samples/Total Samples)	
	0 h	2 h
Positive Control	15/15	15/15
Ethanol	7/15	0/15
0.5% Ginger oil	15/15	7/15
1.0% Ginger oil	1/15	0/15
0.5% Garlic oil	13/15	5/15
1.0% Garlic oil	2/15	0/15
0.5% Ginger oil + 0.5% Garlic oil	1/15	0/15
1.0% Ginger oil + 1.0% Garlic oil	0/15	0/15

## Data Availability

Raw data supporting the conclusions of this manuscript will be made available upon request.

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
