# Peer review of "Ability of Garlic and Ginger Oil to Reduce Salmonella in Post-Harvest Poultry"

_animals, 2022, doi:10.3390/ani12212974_

Round 1

Reviewer 1 Report

Comments to the Authors of manuscript number: animals-1974711 entitled “Ability of garlic and ginger oil to reduce Salmonella in post-harvest poultry.”.

 1. Both simple summary and abstract are written in very good manner, consistent and informative.

2. the introduction explains the source of the contamination with Salmonella, and ways by which infection is transmitted; as well as the manner how it is controlled during food production in poultry industry.

3. the introduction includes also the main information about garlic and ginger oil, which can be used in the prevention of Salmonella infection

4. However, there in the introduction is the lack of clearly pointed out the goal of the study. It should be added.

5. L 103 the source of the bacteria should be stated. Were they from the bank or slaughterhouse. Or were isolated from the animals kept in farm?

6. L 116 – how the skin samples were get? From slaughterhouse or farm? Were samples free from infection?

7. On the other hand, do Authors think that this method is useful in the industry? how big should be the well serving to wash the whole bird in slaughter or how to change the system if the bird could be splashed with oil?

Author Response

Comments to the Authors of manuscript number: animals-1974711 entitled “Ability of garlic and ginger oil to reduce Salmonella in post-harvest poultry.”.

Comment 1:  Both simple summary and abstract are written in very good manner, consistent and informative.

Comment 2. The introduction explains the source of the contamination with Salmonella, and ways by which infection is transmitted; as well as the manner how it is controlled during food production in poultry industry.

Comment 3. The introduction includes also the main information about garlic and ginger oil, which can be used in the prevention of Salmonella infection

Response: We appreciate the thorough review and overall positive feedback from the reviewer.

Comment 4. However, there in the introduction is the lack of clearly pointed out the goal of the study. It should be added.

Response 4: We added a sentence to clarify (see lines 98 to 102).

Comment 5. L 103: The source of the bacteria should be stated. Were they from the bank or slaughterhouse. Or were isolated from the animals kept in farm?

Response 5: All the strains used in the current study were obtained from the culture collection repositories such as American Type Culture Collection (ATCC) or the National Collection of Type Cultures (NCTC). Specifically, S. Infantis BAA 1675, S. Typhimurium ATCC 14028 NA resistant were obtained from ATCC whereas S. Enteritidis PT4 NCTC 13349 NA resistant was obtained from NCTC.

Comment 6. L 116 – How the skin samples were obtained? From slaughterhouse or farm? Were samples free from infection?

Response 6: The skin samples were obtained from the chicken thighs procured from a local grocery store. The skin samples were exposed to UV light for 15 minutes prior to spot inoculation with Salmonella, and we randomly selected 5 skin samples (negative control) and plated the samples. The Salmonella counts were below detection limits in the uninoculated skin samples, and no Salmonella was detected in the enriched samples (data not shown). Further clarification on the source and methods is provided in lines 117 to 119.

Comment 7. On the other hand, do Authors think that this method is useful in the industry? how big should be the well serving to wash the whole bird in slaughter or how to change the system if the bird could be splashed with oil?

Response 7: Thank you for the question and we do believe this method to be useful to the industry. Scalding tanks are a critical part of poultry processing as each carcass is fully submersed in the tank to loosen feathers. As the tanks already exist, the oil could be added directly to the tank without needing to change the system.

Reviewer 2 Report

The authors investigated the antibacterial effect of garlic and ginger oils individually and in combination to inhibit the ability of Salmonella to adhere to host cells, inhibitory role in scalding tank and possible effect on some virulence and vital genes

Abstract

Line 26: please do not specify number; it is better if you give just range or estimation

Line 31: Can you explain why you used peracetic acid other than costumer desire

Line 32-33: Please remove the extra sentence beginning with ‘therefore’ and ending with ‘environment’.

Introduction

Line 47: No need to specify number; just add estimation or range or average

Line 61: What do you mean by vector control?

Line 73: It is not clear why you use Eimeria in you review. A lot of references related to antimicrobial effect against garlic and ginger oil are available. Please change with proper references.

Line 84-89: These sentences need to be merged and rearranged without duplication of references.

Line 94: with many what, is it typo?

Line 96: What is the difference between damage and kill?

Materials and Methods

Line 105: NA, is abbreviation of what? Is it not applicable?

Line 111: What do you mean by desired concentration, please clarify

Line 113: It seems the experiment was conducted only against S. infantis, please clarify that in heading

Line 112: What do you mean by appropriate media? Please clarify

Line 134-144: Sometimes you used abbreviation of minutes and hours and sometime did not; please be consistent.

Line 156-158: It is the same processing for both bacteria, please integrate in one sentence

Table 1: Please reconstruct.

·         Arrange the amplicon size in separate column, replace primer column with F and R letter just beside the primer’s sequences

·         Sequence 5-3 in heading change the punctuation with the right one

·         Left align the sequence of primers

Results:

Line 191-193: The first two sentences can be taken to introduction

Line 202-205: Please write S. infantis instead of salmonella in this part of experiment

Figure 1: Summarize the caption

Table 2: please write treatment solutions instead of treatment in heading

Figure 2: Diagram captions are difficult to read, resolution should be improved, if possible

Figure 4: Summarize the caption, it has a lot of explanation that can be added to the text

Discussion:

Why peracetic acid has some consideration in use?

What do you mean by environmentally friendly?

Why the experiment was limited to S. infantis only?

What were the challenges you faced during experiment performance and how you solved it?

You illustrated positive effect of garlic and ginger in Salmonella reduction, what could be the reasons?

Line 301- 306: These sentences are just repetition of the obtained results that are already illustrated in the results section

Line 321: What can be the reason for less effectivity of aqueous garlic solution?

Line 331: What could be the reason for effectivity of ginger oil only on S. Enteritidis and S. Typhimurium and not on S. Infantis

Line 337: What is your hypothesis of garlic oil effect on invA in S. Enteritidis,

Author Response

The authors investigated the antibacterial effect of garlic and ginger oils individually and in combination to inhibit the ability of Salmonella to adhere to host cells, inhibitory role in scalding tank and possible effect on some virulence and vital genes

Abstract

Comment 1. Line 26: please do not specify number; it is better if you give just range or estimation

Response 1: The line has been changed to increase generality and now says “over 26,000 hospitalizations and 400 deaths”.

Comment 2. Line 31: Can you explain why you used peracetic acid other than costumer desire

Response 2: Peracetic acid is the industry standard as it is effective in reducing bacteria on chicken skin and in the tanks. While customers prefer more natural solutions over chemicals, peracetic acid is still used because there is currently no effective, natural alternative available.

Comment 3. Line 32-33: Please remove the extra sentence beginning with ‘therefore’ and ending with ‘environment’.

Response 3: The sentence has been modified to clearly state the objective of the study.

Introduction

Comment 4. Line 47: No need to specify number; just add estimation or range or average

Response 4: The line has been updated as requested.

Comment 5. Line 61: What do you mean by vector control?

Response 5: Vector control refers to the control of agents that could serve as a vector for Salmonella such as common pests. The line has been updated to “fomite and vector control” to increase clarity.

Comment 6. Line 73: It is not clear why you use Eimeria in you review. A lot of references related to antimicrobial effect against garlic and ginger oil are available. Please change with proper references.

Response 6, Line 73: We modified the sentence to avoid confusion.

Comment 7. Line 84-89: These sentences need to be merged and rearranged without duplication of references.

Response 7: The sentences have been updated.  

Comment 8. Line 94: with many what, is it typo?

Response 8: The phrase “with many” refers to the bacterial strains mentioned immediately before. While the original version is grammatically correct, the line has been updated for clarity.

Comment 9. Line 96: What is the difference between damage and kill?

Response 9: Treatments given at certain levels (such as the sub-inhibitory level used in this paper and the paper cited in line 96) are capable of damaging bacterial cell components (e.g. cell wall, transcription machinery, etc.) without killing the bacteria.

Materials and Methods

Comment 10. Line 105: NA, is abbreviation of what? Is it not applicable?

Response 10: NA stands for Nalidixic Acid. The full name has been added to the line for clarity.

Comment 11. Line 111: What do you mean by desired concentration, please clarify

Response 11: The concentration of oils used in each experiment varies within the study. The sentence has been updated to “…to obtain the desired concentrations for each subsequent experiment.” The concentration used in each experiment is thoroughly explained elsewhere in the methods and results.

Comment 12. Line 113: It seems the experiment was conducted only against S. infantis, please clarify that in heading

Response 12. As requested by the reviewer we add changes “Salmonella” for “S. Infantis” in the heading.

Comment 13. Line 112: What do you mean by appropriate media? Please clarify

Response 13. We modified “appropriate media” to “TSB” to clarify the media used in the study.

Comment 14. Line 134-144: Sometimes you used abbreviation of minutes and hours and sometime did not; please be consistent.

Response 14: We changed the abbreviations to make it consistent throughout the manuscript.

Comment 15. Line 156-158: It is the same processing for both bacteria, please integrate in one sentence.

Response 15. Both bacteria were cultured in the same media. However, V. harveyi strain BB152 was cultured at 30°C for 16 hours or overnight, and the strain V. harveyi strain BB152 was cultured at 30°C for 24 hours before the experiment, because of that, we would like to keep the details of the strains separated in the methods section.

Comment 16. Table 1: Please reconstruct.

  • Arrange the amplicon size in separate column, replace primer column with F and R letter just beside the primer’s sequences
  • Sequence 5-3 in heading change the punctuation with the right one
  • Left align the sequence of primers

Response 16: The column of product size was added to the table, and minor modifications were done to the manuscript.

Results:

Comment 17. Line 191-193: The first two sentences can be taken to introduction

Response 17: The lines have been revised into one sentence that is more appropriate for a results section. 

Comment 18. Line 202-205: Please write S. infantis instead of salmonella in this part of experiment

Response 18: “Salmonella” was substituted for “S. Infantis” as suggested by the reviewer in this results section.

Comment 19. Figure 1: Summarize the caption

Response 19: The caption has been revised.

Comment 20. Table 2: please write treatment solutions instead of treatment in heading.

Response 20: The requested modification was addressed in the manuscript.

Comment 21. Figure 2: Diagram captions are difficult to read, resolution should be improved, if possible

Response 21: The figure has been updated to increase clarity.

Comment 22. Figure 4: Summarize the caption, it has a lot of explanation that can be added to the text

Response 22: The caption has been shortened while still seeking to retain all of the information necessary for the figure to stand alone.

Discussion:

Comment 23. Why peracetic acid has some consideration in use?

Response 23: Peracetic acid (PAA) is commonly used by the industry to reduce bacteria on carcasses and in the environment during poultry processing. The mention of peracetic acid on line 288 is to provide a rationale for the study.

Comment 24. What do you mean by environmentally friendly?

Response 24: By environmentally friendly, we mean that these are naturally occurring products that are not considered to have a negative effect on the environment.

Comment 25. Why the experiment was limited to S. infantis only?

Response 25: S. infantis was used as a representative strain in the chicken skin experiment.

Comment 26. What were the challenges you faced during experiment performance and how you solved it?

Response 26: Multiple challenges were faced during this experiment. Of note was the process of determining the best incubation time and phytochemical dose for the first experiment. Investigation into industry standards and preliminary experiments helped greatly to overcome these challenges.

Comment 27. You illustrated positive effect of garlic and ginger in Salmonella reduction, what could be the reasons?

Response 27: There could be multiple modes of action that we were not able to explore in this paper. For example, the oils could damage the cell membrane resulting in bacterial death.

Comment 28. Line 301- 306: These sentences are just repetition of the obtained results that are already illustrated in the results section

Response 28: The reviewer’s comment is noted, the first line has been removed, but the second has been kept to facilitate the discussion that follows well.

Comment 29. Line 321: What can be the reason for less effectivity of aqueous garlic solution?

Response 29: The authors of the cited articles indicate that the difference may be due to decreased stability of the aqueous garlic solution.

Comment 30. Line 331: What could be the reason for effectivity of ginger oil only on S. Enteritidis and S. Typhimurium and not on S. Infantis

Response 30. Line 331:  Ginger oil affected all three serovars of Salmonella, but S. Infantis has less than the other two. We have rephrased the sentence.

 Comment 31. Line 337: What is your hypothesis of garlic oil effect on invA in S. Enteritidis,

Response 31. Line 337: We are planning to perform an in-depth study to identify why it affects Enteritidis, not the others. 

Reviewer 3 Report

The manuscript presents interesting studies to reduce Salmonella in post-harvest poultry. However, there are some issues that it is important for authors to address.

L.61. vector control, aim to prevent...

L.107. In which phase of the bacterial growth kinetics were the inocula used to carry out the antimicrobial activity studies? How would you demonstrate in which phase of bacterial growth the different strains of Salmonella are found?

L.117. Figure 1 only shows 9 treatments, indicate if one more is missing or correct it, please.

L.125. What was the actual concentration? It is important to know it to adjust the corresponding calculations.

L.128. How would you guarantee that Salmonella is resistant at 54 ºC? This is a somewhat high temperature, right?

L.171. At what time is the logarithmic mean phase reached?

L.194. independent experiments.

L.195-196. As mentioned above, what is the effect of temperature on bacterial viability? How are oils affected at 56 ºC in terms of composition? Were the composition studies of the oils carried out before and after being subjected to 56 ºC? 

L.198-199. The experiments show that there is a decrease in the bacterial concentration, but how representative are the studies at 56 ºC? Is the use of these oils feasible in economic terms?

L.204-205. Are there significant differences between bacterial concentration (0.8 vs. 1.2 log CFU) when contact time was compared? This is important because perhaps there are no significant differences.

Figure 1. Statistical analysis was performed considering all the data from the 3 independent experiments (n=9/group)? It is suggested to place the results independently in tables marking their significance with superscripts. This is because it is not observed if there were differences between treatment 8 and 9.

It is logical that the combination of 1% of each of the oils would have better antimicrobial activity since in reality a concentration of 2% of a mixture of oils is being tested.

It is logical that the combination of 1% of each of the oils would have better antimicrobial activity since in reality a concentration of 2% of a mixture of oils is being tested.

Looking at the results for 0.5% of each oil (1% total), there were no significant differences when compared to 1% of the individual oils.

Table 2. This result suggests that ethanol had the antimicrobial effect or or simply the temperature of 56 ºC.

L.238-239. Was this apparent reduction in bacterial growth significant?

L.304-306. This is not logical since the initial concentration was around 6, in any case it would be reducing 1/3 of bacterial concentration. 

Author Response

The manuscript presents interesting studies to reduce Salmonella in post-harvest poultry. However, there are some issues that it is important for authors to address.

Comment 1. L.61. vector control, aim to prevent...

Response 1: Vector control aims to prevent the introduction/transmission of pathogens into the hatchery environment via vectors such as rodents, insects, or objects such as worker garments.

Comment 2. L.107. In which phase of the bacterial growth kinetics were the inocula used to carry out the antimicrobial activity studies? How would you demonstrate in which phase of bacterial growth the different strains of Salmonella are found?

Response 2: We used Salmonella in mid-log phase for inoculation. The bacterial growth kinetics were determined based on the previously generated growth curves for the select Salmonella strains and also from the SIC data generated using the Cytation 5.

Comment 3. L.117. Figure 1 only shows 9 treatments, indicate if one more is missing or correct it, please.

Response 3: The authors are unsure of what the reviewer is referring to. The nine treatments shown in the figure are the only treatments. They are the same as those listed in the materials and methods (Lines 120 to 123) and the results (Section 3.1.).

Comment 4. L.125. What was the actual concentration? It is important to know it to adjust the corresponding calculations.

Response 4: The concentration of ethanol for this treatment was 9%. This information was added to the manuscript (Line 125).

Comment 5. L.128. How would you guarantee that Salmonella is resistant at 54 ºC? This is a somewhat high temperature, right?

Response 5: We included appropriate controls, including a baseline and untreated Salmonella positive control for comparison. We also conducted a preliminary study to ensure that the Salmonella can withstand the scalder tank temperatures (data not shown).

Comment 6. L.171. At what time is the logarithmic mean phase reached?

Response 6: Approximately 8-9 hours

Comment 7. L.194. independent experiments.

Response 7: The sentence has been updated as suggested.

Comment 8. L.195-196. As mentioned above, what is the effect of temperature on bacterial viability? How are oils affected at 56 ºC in terms of composition? Were the composition studies of the oils carried out before and after being subjected to 56 ºC? 

Response 8: As mentioned above, the temperature did not have an effect on bacterial viability, as shown by the baseline and positive control samples. A sentence has been added (Lines 203 – 204) for clarity. However, studies on the effect of scalding tank temperature on the composition of the oils were not carried out.  

Comment 9. L.198-199. The experiments show that there is a decrease in the bacterial concentration, but how representative are the studies at 56 ºC? Is the use of these oils feasible in economic terms?

Response 9: Scalder tank is one of the critical areas in poultry processing where cross-contamination may occur. Intervention strategies during different poultry processing steps, specifically during scalding may be ideal as all the birds will be dipped into the same scalder tank which in turn increases the chances for cross-contamination. Since scalding is one of the first steps in processing, antimicrobial interventions at this stage provide a practical and effective means of reducing pathogen load and can minimize the risk of cross-contamination of carcasses further down the processing steps.

Comment 10. L.204-205. Are there significant differences between bacterial concentration (0.8 vs. 1.2 log CFU) when contact time was compared? This is important because perhaps there are no significant differences.

Response 10: Yes, the difference between contact times was investigated, and it found that the bacterial reduction in the 1% garlic and ginger combination was significantly different between times, while the 0.5% combination was not. The information has been added to lines 213 to 215.

Comment 11. Figure 1. Statistical analysis was performed considering all the data from the 3 independent experiments (n=9/group)? It is suggested to place the results independently in tables marking their significance with superscripts. This is because it is not observed if there were differences between treatment 8 and 9.

Response 11: While we appreciate the reviewer's suggestions, this paper focuses on treatments' ability to reduce Salmonella versus the untreated positive control. Therefore, the figure has been left in its original state.

Comment 12: It is logical that the combination of 1% of each of the oils would have better antimicrobial activity since in reality a concentration of 2% of a mixture of oils is being tested.

Response 12: We agree with the reviewer’s conclusion.

Comment 13: It is logical that the combination of 1% of each of the oils would have better antimicrobial activity since in reality a concentration of 2% of a mixture of oils is being tested.

Response 13: We agree with the reviewer’s conclusion.

Comment 14: Looking at the results for 0.5% of each oil (1% total), there were no significant differences when compared to 1% of the individual oils.

Response 14: For both time points, the 1% of individual oils is not significantly different from the positive control (BPD alone). As the goal of this study was to find treatments significantly different from simply dipping in water or BPD, whether they are significantly different from the combination treatments is irrelevant.

Comment 15: Table 2. This result suggests that ethanol had the antimicrobial effect or simply the temperature of 56 ºC.

Response 15: The lack of bacterial loss in the positive control solutions (BPD solution in which contaminated samples were dipped) indicates that the temperature alone is not responsible for the antimicrobial effect. Furthermore, the lack of antimicrobial efficacy of ethanol alone on chicken skin (figure 1) and a greater decrease in positive solutions in the presence of oils (table 2) would indicate that the antimicrobial effect is due to the oils.

Comment 16: L.238-239. Was this apparent reduction in bacterial growth significant?

Response 16: Statistics were not run on this data as it only measures the presence and absence of growth in undiluted solutions rather than plate counts.

Comment 17: L.304-306. This is not logical since the initial concentration was around 6, in any case it would be reducing 1/3 of bacterial concentration. 

Comment 17: L.304-306: Sorry for the confusion. We are referring to 1 log or 2 log reduction, not a reduction from 6 log CFU to 2 log CFU. Since a 50% reduction in the prevalence of Salmonella-contaminated chickens can reduce human risk by 50%; 1 log reduction in Salmonella can reduce human risk by 90%, and 2 log reduction can reduce the risk by 99%, which is significant. We revised the sentence for clarity.

Reviewer 4 Report

Garlic and ginger oils have proven antimicrobial activity and possess many benefits for human health. In this study, the ability of garlic and ginger to reduce Salmonella in the processing environment. The manuscript fall in the the topic of toxins journal and was interesting for the readers. It was well prepared. I suggested it could be accepted in the present form.

Author Response

Comment 1: Garlic and ginger oils have proven antimicrobial activity and possess many benefits for human health. In this study, the ability of garlic and ginger to reduce Salmonella in the processing environment. The manuscript fall in the the topic of toxins journal and was interesting for the readers. It was well prepared. I suggested it could be accepted in the present form.

Response 1: We greatly appreciate the reviewer’s favorable response.